

# Drift corrected Odin-OSIRIS ozone product: algorithm and updated stratospheric ozone trends

Adam E. Bourassa[1], Chris Z. Roth[1], Daniel J. Zawada[1], Landon A. Rieger[1], Chris A. McLinden[2], Douglas A. Degenstein[1]

[1]Institute of Space and Atmospheric Studies, University of Saskatchewan, Saskatoon, Canada
[2]Environment and Climate Change Canada, Downsview, Ontario, Canada

*Correspondence to*: Adam E. Bourassa (adam.bourassa@usask.ca)

**Abstract.** A small, long-term drift in the Optical Spectrograph and Infrared Imager System (OSIRIS) stratospheric ozone product, manifested mostly since 2012, is quantified and attributed to a changing bias in the limb pointing knowledge of the

instrument. A correction to this pointing drift using a predictable shape in the measured limb radiance profile is implemented and applied within the OSIRIS retrieval algorithm. This new data product, version 5.10, displays substantially better both long- and short-term agreement with MLS ozone throughout the stratosphere due to the pointing correction. Previously reported stratospheric ozone trends over the time period 1984-2013, which were derived by merging the altitude/number density ozone profile measurements from the Stratospheric Aerosol and Gas Experiment (SAGE) II satellite instrument

(1984–2005) and from OSIRIS (2002–2013) are recalculated using the new OSIRIS version 5.10 product, and extended to 2017. These results still show statistically significant positive trends throughout the upper stratosphere since 1997, but at weaker levels that are more closely in line with estimates from other data records.

## 1 Introduction

Satellite observations have been used for over three decades to measure, diagnose and ultimately monitor stratospheric ozone

variability and long term trends. Limb profiling observations have been of particular interest and importance, and analyses of these measurements have led to height resolved quantification of long term trends in stratospheric ozone and the effect of various drivers of variability including the quasi-biennial oscillation, the solar cycle, and the El Nino Southern Oscillation (for example see Weatherhead et al., 2000, Randel and Wu, 2007; Kyrölä et al., 2013, Bourassa et al., 2014; Harris et al., 2015).


The most recent World Meteorological Organization (WMO) Scientific Assessment on Ozone Depletion (WMO, 2014) used many satellite based limb data sets. Two studies published shortly after (Harris et al., 2015; Tummon et al., 2015) also used these satellite limb data sets. All three works concluded that the statistically significant ozone depletion occurring before approximately the year 2000 was not evident in the post-2000 period. However, the studies diverged on the existence of a

statistically significant recovery. Harris et al. (2015) state that "there is a hint of an average positive trend of approximately



2% per decade in mid-latitudes and approximately 3% per decade in the tropics" while WMO, 2014 conclude that there was a statistically significant increase in upper stratospheric of between 2.5 and 5 percent per decade in the post 2000 timeframe. Tummon et al. (2015) wrote that there was a statistically significant increase in upper stratospheric ozone in the analysis of certain data records but not all. They do conclude, however, that "the mean trend of all data sets is significant and positive in

the southern midlatitudes between 5 and 0.5 hPa, in the tropics between 3 and 1 hPa, and in the northern mid-latitudes at just 2 hPa".  In light of these apparently conflicting results as Harris et al. (2015) stated, "the significance will become clearer as (i) more years are added to the observational record, (ii) further improvements are made to the historic ozone record (e.g. through algorithm development), and (iii) the data merging techniques are refined, particularly through a more rigorous treatment of uncertainties.". These three points serve as goals of the new SPARC activity Long-term Ozone Trends and

Uncertainties in the Stratosphere (LOTUS) (http://www.sparc-climate.org/activities/ozone-trends/).

The original ozone trend work using the Odin-OSIRIS instrument (Murtagh et al., 2002; Llewellyn et al., 2004; McLinden et al., 2012) ozone data was performed by Sioris et al., 2014, and focused only on the tropical lower stratosphere.  This was extended by Bourassa et al., 2014 to cover altitudes from 18 km to 50 km over the latitude range 60°S to 60°N.  Both of

these studies performed a merging of the OSIRIS version 5.07 ozone data product, from 2002-2013, with the version 7 solar occultation profiles obtained by the Stratospheric Aerosol and Gas (SAGE) II satellite instrument, which extend over 1984-2005 (McCormick et al., 1989; Damadeo et al., 2013).  The OSIRIS limb scattered sunlight measurements have precision and resolution similar to the SAGE II measurements, and merge in a straightforward fashion due to four years of mission overlap.  The merging is also simplified by the fact that for both instruments the inherent product is ozone number density on

altitude levels such that complications from conversions from mixing ratio on pressure surfaces do not arise (McLinden and Fioletov, 2011).  The conclusion of Bourassa et al. 2014 was that from 1997 to 2013 (the end of the OSIRIS data set analyzed at that point) there was a statistically significant recovery of 3-8% per decade throughout the stratosphere, except in the tropical lower stratosphere, where small but significant negative trends persisted.

It was noted by Bourassa et al., 2014, that the derived positive trends were somewhat higher than those reported from a similar merging analysis by Kyrola et al., 2013, using the SAGE II and Global Ozone Monitoring by Occultation of Stars (GOMOS) measurements.  They also noted that the long-term stability of OSIRIS up to 2012, was assessed by Adams et al. (2014) using comparisons between OSIRIS, GOMOS, the Microwave Limb Sounder (MLS) and ozonesondes. This analysis showed that the OSIRIS data product did show some regions of positive drift (meaning higher ozone values) with respect to

the other data sets, particularly after 2010; however, the global average drifts were < 3% per decade between approximately 18-45 km.  Bourassa et al., 2014 chose not to modify the calculated trends to account for the possible drift, and instead, modified the statistical uncertainty by adding 3% per decade in quadrature to the calculated uncertainties from the regression.



More recently, the bias and drift of a large number of satellite limb profile ozone data records, many of which were used for trend estimation, were systematically assessed by Hubert et al., 2016. The stability of each of the limb data records was tested against ground-based ozonesondes and a stratospheric ozone lidar network. This study found that the drift against these ground-based measurements for most limb instruments is smaller than ±5% per decade; however, a small number of
the instruments, including OSIRIS, were found to have somewhat higher, significant, long term drifts. The summary concludes that a small positive drift is present in OSIRIS ozone between 20-35 km (smaller than, or close to the defined 5% significance threshold), and a larger drift (up to 8% per decade in some regions) exists at upper stratospheric altitudes. In the most recent years, i.e. since 2014, this drift has become larger and visually apparent in the ozone time series. For example, Fig. 1 shows the OSIRIS version 5.07 ozone time series at 40.5 km, for latitude bins centred at 30 S and 30 N, as well as the
MLS ozone time series at the same locations. The reported MLS volume mixing ratio profiles on pressure levels are converted to number density on altitude levels using ECMWF ERA-Interim reanalysis fields (Dee et al., 2011). The vertical dashed line on Fig. 1 shows the end of the data set used for the Bourassa et al., 2014 trend analysis. It is clear that since approximately that point there is a definitive positive drift in the OSIRIS measurements with respect to MLS. There is a smaller, but also noticeable, positive drift starting as early as 2010.

In this work, we identify the source of the drift in the OSIRIS ozone product as a small systematic error in the limb pointing knowledge of the instrument that is nominally determined by on-board star trackers. This error has a seasonal cycle and also has slowly increased in magnitude since approximately 2010. The pointing error is estimated using a variant of a well-known technique for limb scatter altitude registration using the shape of the UV radiance profile. We have implemented this
pointing correction into the tangent point registration of the limb radiance profiles and re-processed the entire OSIRIS ozone data product, released as version 5.10, up to the present time using the identical algorithm from the previous version 5.07 (Degenstein et al., 2009) that was used for the trend assessments. The effectiveness of the pointing correction is demonstrated through comparison of the time series with MLS measurements, and revised SAGE II-OSIRIS trend results are presented showing recovery throughout the upper stratosphere that is still statistically significant, but at weaker levels than
previously reported.

## 2  Pointing Correction Technique

The Rayleigh Scattering Attitude Sensor (RSAS) is an instrument that measured the characteristic shape of the limb scattered sunlight radiance profile from Space Shuttle STS-72 (Janz et al., 1996) in order to determine the altitude registration of the tangent points of the instrument lines of sight. This approach, now generally referred to in the community as the RSAS
technique, has since been tested and/or used in several variants for many limb scatter measurements including those from the Shuttle Ozone Limb Sounding Experiment (SOLSE) (McPeters et al., 2000), the Limb Ozone Retrieval Experiment (LORE)



(Flittner et al., 2004), SAGE III (Rault, 2005), more recently for the Ozone Mapping and Profiler Suite (OMPS) (Moy et al., 2017).

Briefly, the RSAS technique takes advantage of the fact that the shape of the limb radiance at 350 nm has a characteristic
"knee" in the profile near 20 km due to the high optical thickness of the limb line-of-sight. Thus, near and below this altitude the limb radiance is roughly constant and essentially insensitive to the limb pointing. Alternately, above 30 km the atmospheric path is quite optically thin and the limb radiance approximately follows the exponential pressure gradient making it very sensitive to the altitude of the tangent point. Therefore the ratio of radiances from these two altitude regions can be used to compare to a model calculation of the same quantity to test the altitude registration of the profile. The ratio
provides a cancellation of the effect of the unknown diffuse upwelling radiance from clouds and the Earth's surface, and removes the requirement for an absolute calibration of the instrument. Also, this wavelength has essentially zero ozone absorption; however, the technique does have some sensitivity to stratospheric aerosol that can bias the results. See Moy et al., 2017, for a more detailed discussion of the general RSAS technique.

We have used a simple implementation of the RSAS technique for the OSIRIS pointing correction. We define the RSAS ratio for a limb scan as the mean of the logarithm of the measured 350 nm limb radiance for all tangent points falling between 18.5-22.5 km divided by the same for tangent points between 30.5-36.5 km. This same ratio is then calculated with the SASKTRAN radiative transfer model (Bourassa et al., 2008; Zawada et al., 2016) by simulating the limb radiance at the same tangent points as the measurement. The difference between measured and modelled ratios is minimized in a least
squares sense by shifting the tangent points of the measured radiance profile by interpolating the logarithm of the measured radiance in altitude.

The radiative transfer calculation is done using the version 5.07 retrieved ozone and stratospheric aerosol profiles as inputs. Temperature and air density are taken from ECMWF ERA-Interim (Dee et al., 2011). Before the RSAS calculation, the
albedo is first retrieved for each scan at 350 km by adjusting the albedo in the radiative transfer model such that measured and modelled radiance match at 40 km tangent altitude. The RSAS shift for any given OSIRIS limb scan is not calculated and the scan is excluded from the analysis if the retrieved aerosol extinction above 18 km is greater than $3 \times 10^{-4}$ km$^{-1}$, which is approximately the level that starts to cause uncertainty in RSAS greater than the inherent measurement noise.

The calculated tangent height shifts for every applicable scan in the entire OSIRIS mission are shown with blue markers in Fig. 2, and the daily average shift is indicated in black. The RSAS analysis shows that a relatively strong annual cycle in the pointing correction on the order of 400 m beginning in April/May of every year when the Odin satellite enters solar eclipse for a fraction of every orbit. During this period, the eclipse geometry causes stronger thermal cycles on the satellite than during the rest of the year. For example, the OSIRIS optics box temperature is also shown on Fig. 2 and indicates the





colder temperatures observed during the summertime eclipse period. Note that the RSAS shift is positively correlated with this temperature. It is likely that these colder temperatures cause thermal contraction that results in an angular offset of the assumed static boresight of OSIRIS with respect to the Odin reference frame and therefore a limb pointing offset.

5  An upward trend in the calculated RSAS shifts is also observed in the later years of the mission, i.e. since approximately 2012, which qualitatively matches the observed drift in OSIRIS ozone shown in Fig. 1. Interestingly the temperature of the optics box also has a trend in the later years of the mission; however, it begins earlier than 2012 and is negatively correlated with the derived RSAS shift. The cause of the thermal drift is related to the slow precession and decay of the Odin orbit and is not well understood. The fact that the sign of the correlation between optics temperature and RSAS shift is opposite with 10  respect to the eclipse period and the long-term thermal drift means that this temperature cannot be used in any predictive sense. Even outside the eclipse period, the correlation is weak (i.e. 0.3 for the months of January). However, we include it here simply to show that the changing thermal environment of the satellite is very likely related to the errors in the pointing knowledge.

15  Prior to ~2011 and outside of the eclipse season the RSAS technique suggests there is a bias in the pointing offset on the order of 200 m. So to better understand the limitations of the RSAS technique we analyse the calculated tangent height offsets between 2004 and 2010 and outside of the eclipse period. Previous studies have detected no time dependent biases in the OSIRIS measurements outside of the eclipse periods for these years, and have also observed no meridional dependence in high altitude ozone biases (Adams et al., 2013; Adams et al., 2014). Therefore, we would expect the average tangent 20  height offset to be zero with no latitudinal structure during this time. However, when we bin all of the RSAS values in latitude and longitude outside of the eclipse time period, there are distinct geographic features with an average offset of ~200 m. This is shown on the map in Fig. 3. The geographic structure is reminiscent of high altitude cloud distributions and likely indicates a deficiency in the assumption of a Lambertian surface for modelling the diffuse upwelling radiation in the presence of clouds.

25

If this observed spatial structure is indeed from a deficiency in the modelling and not part of the true pointing offset of the OSIRIS instrument, we can perform a first order correction for this structure by correcting the derived RSAS shift by the geographic average shown in Fig. 3 as a type of calibration for the technique. Thus for every scan's calculated RSAS offset, we subtract the 2004-2010 spatial average, essentially forcing the spatial average value to be zero from 2004 to 2010. These 30  corrected RSAS shifts for each scan are then averaged over each day to create the full mission daily RSAS shift that is shown in Fig. 4. Scans with RSAS shift values greater than 1000 m, which is well outside the normal variability, are rejected in the daily average. Also, regions with high aerosol extinction or high/low albedo values that are discarded by the RSAS algorithm can have a large extent geographically, but even so there are still typically many valid scans in any given day that are used to construct the daily average. The final time series of daily average RSAS shift, shown in Fig. 4, is then



interpolated to the time of every scan to calculate the offset that is applied to the tangent altitudes for the version 5.10 processing.

## 3 The version 5.10 ozone product

The version 5.10 ozone product is produced using the same retrieval algorithm as the previously released version 5.07,

which includes retrievals of albedo, stratospheric aerosol extinction, NO2, and O3. No changes were made to the retrievals of the other species other than the pointing correction to the radiances; the impact of this correction on these other species has yet to be assessed. Note that it is important that the RSAS pointing correction is applied to the limb radiances before the retrieval, as even though the retrieval algorithm is identical, this is not equivalent to simply shifting the retrieved ozone profile by the RSAS correction. The retrieval is non-linear, particularly in the lower stratosphere, and the altitude

registration of the limb radiances must be corrected before the retrieval in order to obtain a self-consistent result.

The only other change from version 5.07 is the change of temperature and air density profiles used in the forward model from ECMWF-operational to ECMWF ERA-Interim. As a global reanalysis, the ERA-Interim product is a highly appropriate choice for this mission reprocessing, as the ~2 month delay in release of the products is not a concern. It has also

been encouraged as a standard input for this type of application by the SPARC SI2N working group (http://igaco-o3.fmi.fi/VDO/). We do not expect this change to create a large difference between the product versions, so in the interest of efficient utilization of computational resource we implemented this change in the same reprocessing as the RSAS pointing correction.

The effectiveness of the RSAS pointing correction in the version 5.10 product is demonstrated in Fig. 5, which shows the percent difference in daily zonal mean ozone relative to MLS v4.2 for both OSIRIS v5.07 and v5.10 in the tropics at 40.5 km. The v5.07 data exhibits a (8.97 ± 0.56) % / decade drift relative to MLS while the RSAS corrected v5.10 shows substantially reduced drift of (0.69 ± 0.26) % / decade. Note that here we simply fit a straight line to the time series and did not attempt to fit and/or correct for any annual or interannual cycles. Other latitudes and altitudes show similar behaviour,

although in many locations the drift in v5.07 was not as large as that shown in this example.

The overall observed variability at shorter time scales has also decreased significantly in the v5.10 comparison. This effect can be observed in most time periods, but it is most noticeable from 2005 to mid-2007. The reduction in variability of this difference on a daily average scale gives us confidence that the short term variability in the calculated RSAS correction is in

fact real and not simply noise or an artefact of the technique.



During the summertime orbital eclipse, differences relative to MLS are also greatly reduced, but still present. It is possible that the RSAS technique is underestimating the necessary offset, but this seems unlikely as the overall drift is corrected nearly perfectly and is of the same order of magnitude. A more likely possibility is that the change in optics temperature during eclipse times has a separate effect on the OSIRIS spectral point-spread function which is not being accounted for in the retrieval process. These differences are still under investigation and will be addressed in future versions of the retrieval algorithm.

## 4 Impact on Trends

To examine the impact of the pointing correction on the derived ozone trends from the merged SAGE II version 7 and OSIRIS records we have used the version 5.10 ozone product to perform the same analysis of Bourassa et al., 2014, now extended to 2017. Briefly, the merged and de-biased anomalies, binned as a function of latitude and altitude are fit with a multivariate linear regression model that includes predictor functions including principal components of the quasi-biennial oscillation, the F10.76 solar flux, the El Nino Southern Oscillation index, the tropopause pressure, and a piecewise linear term with inflection point fixed at 1997. For a detailed development and discussion of the regression model see Bourassa et al., 2014. Example deseasonalized ozone anomalies for the same two latitude/altitude bins shown in Fig. 1, calculated for both OSIRIS v5.10 and MLS v4.2, are shown in Fig. 6. Note that there is no discernible drift between the two instruments and the short term variability is well matched throughout the data records.

With the linear regression model, we find very similar results to Bourassa et al., 2014 in terms of the significance of the predictor functions and the capability of the model to explain the overall variability. The correction of the pointing error did, however, have an impact on the calculated linear trends, mostly since 1997. The magnitude of the linear trends for pre- and post-1997 are shown in Fig. 7, with black contours marking steps of 2% per decade. Cross hatching marks regions that are not statistically significant with the AR1 regression.

The trends found here in the post-1997 period show 1-3% per decade recovery with significance throughout most of the stratosphere above 25 km, except for a region of essentially zero trend extending through most altitudes between 10 S and 10 N and in the Southern Hemisphere between 25 and 35 km. In most locations these trends are smaller than the 3-8% per decade reported by Bourassa et al., 2014. The overall structure is different as well and displays a reduction in the asymmetrically strong Southern Hemisphere recovery found previously. Note that because the latitudinal coverage varies throughout the year due to the sun-synchronous terminator orbit, the daily average RSAS pointing correction can manifest with meridional structure even though latitudinal dependence is averaged over any given day. The decreasing trend in the tropical lower stratosphere, found previously by Sioris et al., 2014 and Bourassa et al., 2014, remains statistically significant in these updated and pointed corrected results, although with slightly reduced extent.





## 5 Summary and Conclusion

The OSIRIS ozone product was one of several satellite limb datasets used in the 2014 WMO ozone assessment and other supporting studies during the same time frame. Overall the conclusions of these studies were varied as to the magnitude and significance of ozone recovery. Trends reported at that time by Bourassa et al., 2014, using merged SAGE II and OSIRIS measurements were between 3-8% per decade throughout most of the stratosphere, which was generally higher than detected by most other instruments.

A positive drift in the OSIRIS ozone measurements, independently identified by Hubert et al., 2016, through comparison with ground-based observations, is attributed here to a systematic error in the pointing knowledge of the OSIRIS limb radiance measurements, affecting the altitude registration of the profile mostly since 2012. A variant of a well-known technique for pointing determination from the shape of the limb radiance profile, called RSAS, is developed and applied to the OSIRIS measurements. The derived correction was then used to generate a new version of the retrieved ozone product, publically released as version 5.10.

Comparisons with MLS ozone show that the drift in version 5.07 is substantially reduced, together with a convincing decrease in the short term differences observed in the two products. The trend calculation of Bourassa et al., 2014, is repeated with the new version 5.10 product and updated to 2017. These results still show statistically significant recovery throughout much of the upper stratosphere, but at reduced levels of 1-3% per decade.





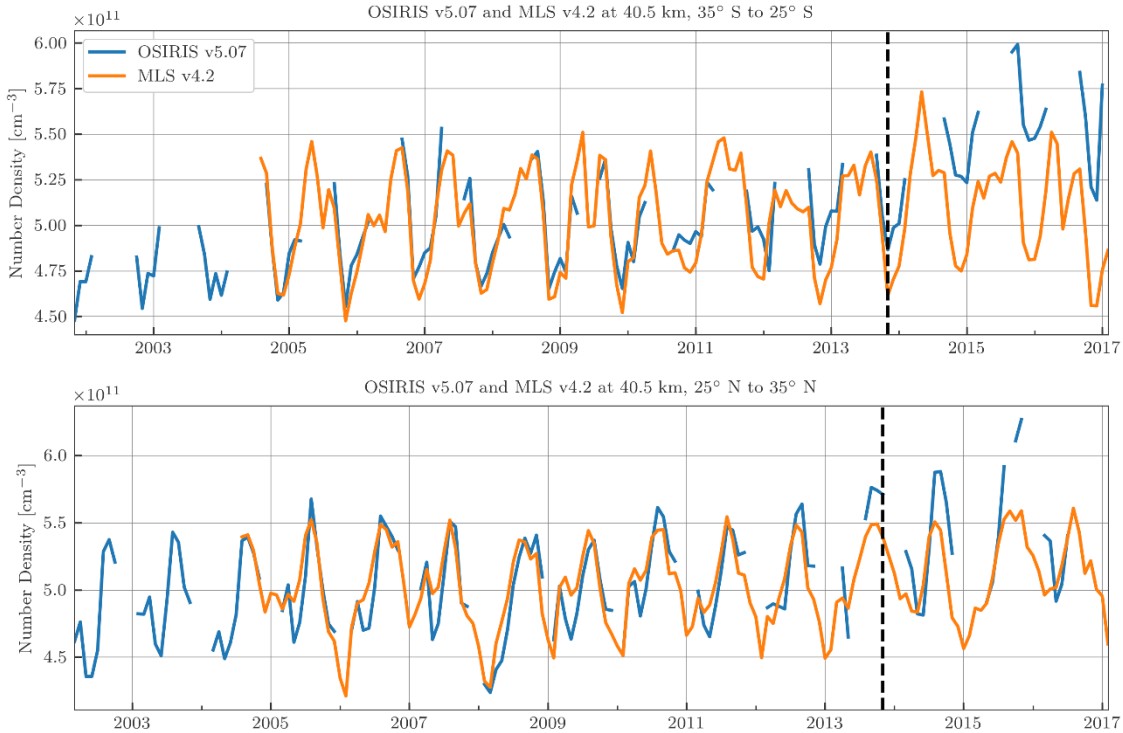

**Figure 1: OSIRIS version 5.07 and MLS version 4.2 ozone density at 40.5 km and 35 S to 25 S (top), and 25 N to 35 N (bottom).**
5    **The end of the OSIRIS data set used in the Bourassa et al., 2014 trend study is marked with the vertical black dashed line.**

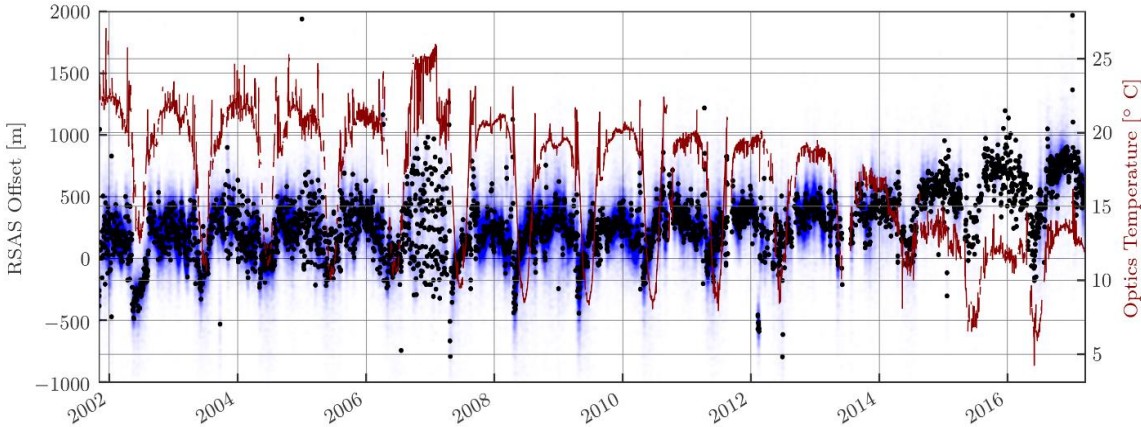

**Figure 2: Calculated RSAS tangent height offset for every OSIRIS scan (transparent blue dots) and daily averages (black dots).**
10    **The daily average OSIRIS optics box temperature is shown as the red line.**



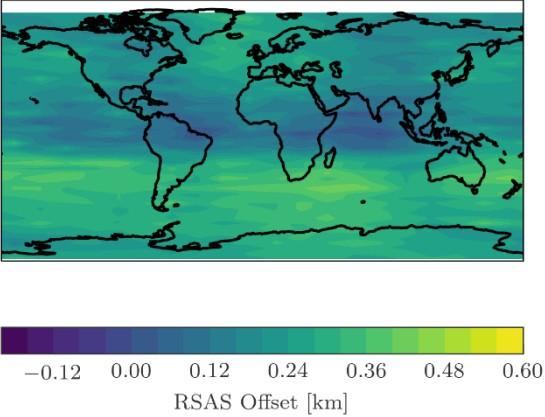

**Figure 3: Globally binned in latitude and longitude RSAS offsets for time period beginning in 2004 and ending in 2010. Values where the OSIRIS optics temperature is less than 18° C are excluded.**

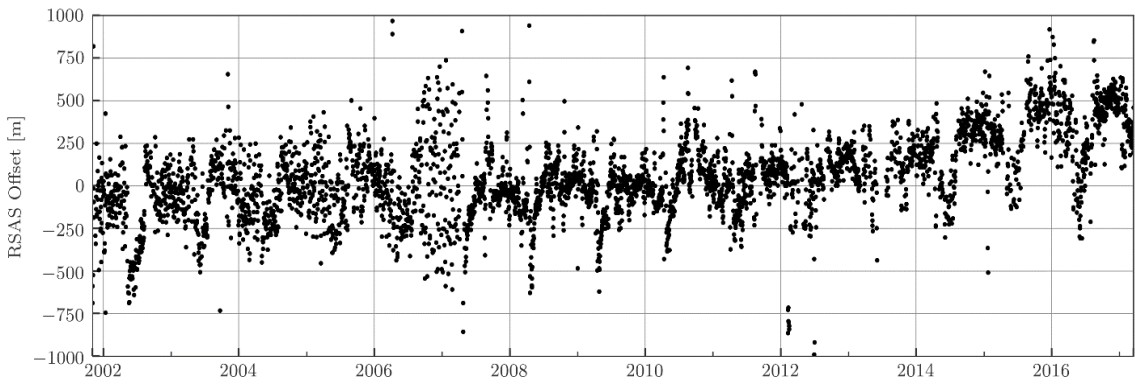

**Figure 4: Daily average RSAS tangent height offset after subtracting the geographic average from GEOGRAPHIC AVERAGE FIGURE. This is the final correction that is used in v5.10 data processing.**

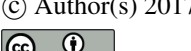



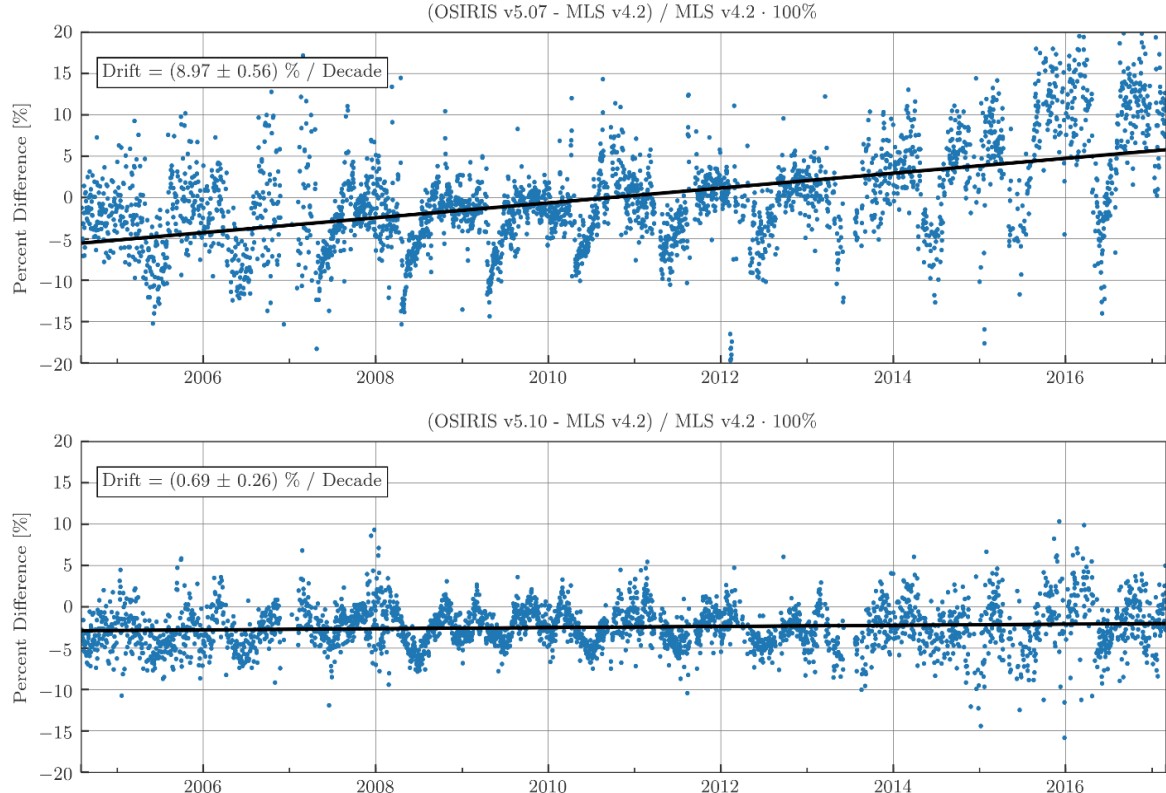

**Figure 5: Daily averaged percent difference in zonal mean ozone from 10° S to 10° N at 40.5 km relative to MLS v4.2 for OSIRIS v5.07 (top) and OSIRIS v5.10 (bottom). The MLS v4.2 data is converted to number density on an altitude grid using ERA interim reanalysis pressure and temperature fields. Simple best fit straight line drift estimates are shown in black with uncertainties reported as the 2σ values.**





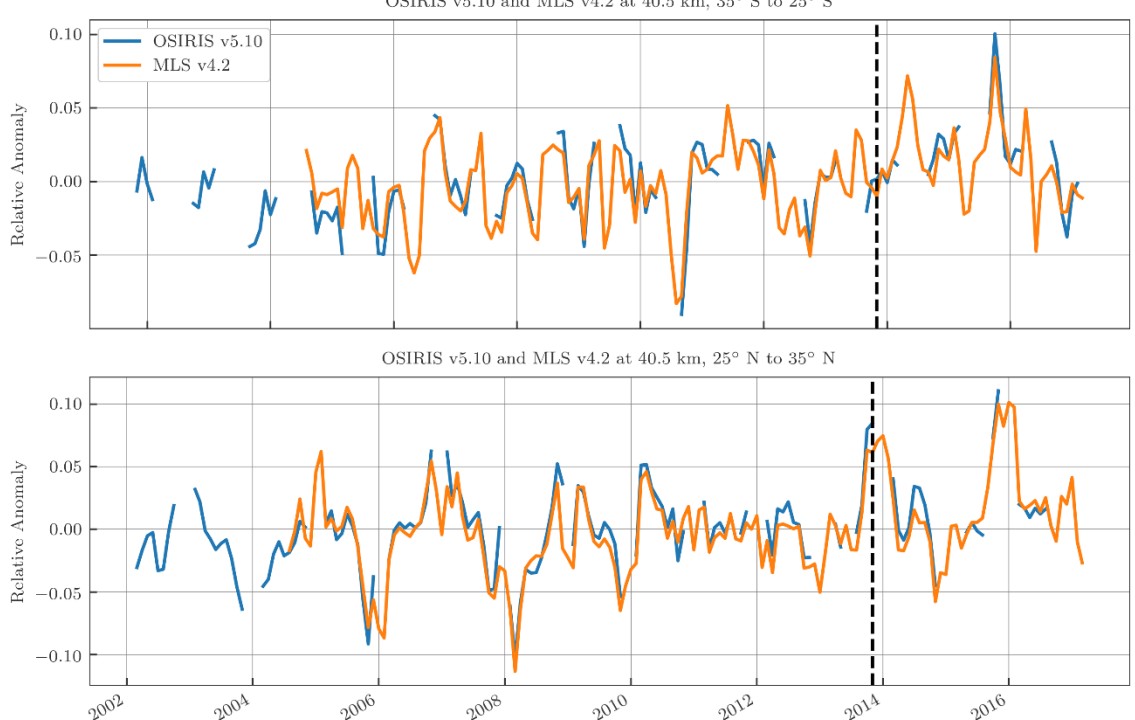

**Figure 6: Ozone anomalies used for the trend calculation from pointing corrected OSIRIS version 5.10, together with the same from MLS at the same locations specified in Figure 1.**

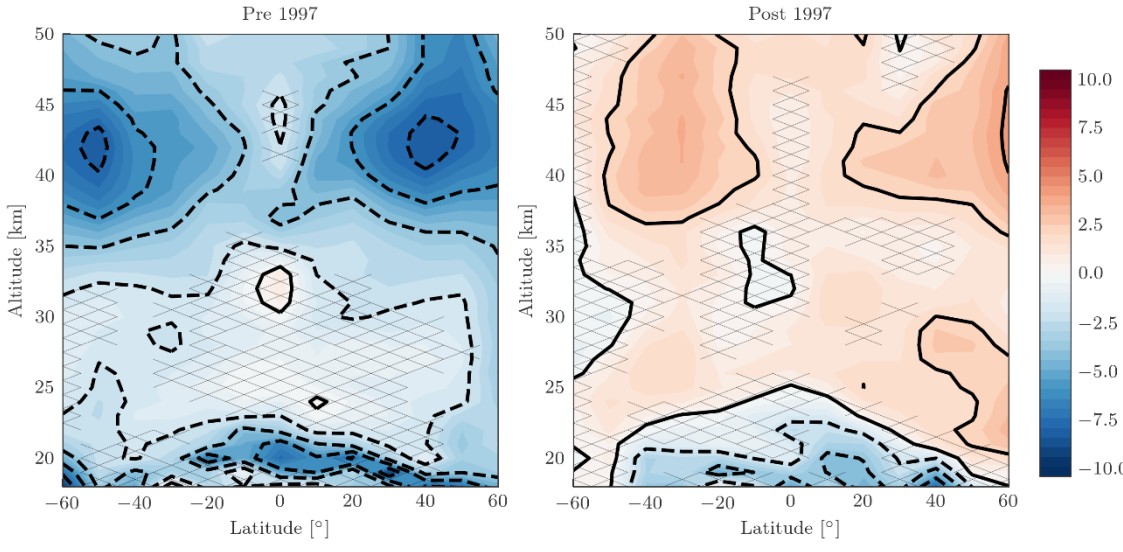

**Figure 7:  Updated linear ozone trends pre- and post-1997, in percent per decade, following the method of Bourassa et al., 2014, using SAGE II version 7 and the pointing corrected OSIRIS version 5.10 ozone product.  The cross-hatching denotes regions that are not statistically significant. Black contours mark steps of 2% per decade.**



## Data Availability

OSIRIS data are available for download at http://odin-osiris.usask.ca/

## Acknowledgements

This work was supported by the Natural Sciences and Engineering Research Council (Canada) and the Canadian Space
Agency (CSA). Odin is a Swedish-led satellite project funded jointly by Sweden (SNSB), Canada (CSA), France (CNES)
and Finland (Tekes).

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
