# Peer review of "Drift corrected Odin-OSIRIS ozone product: algorithm and updated stratospheric ozone trends"

_Atmospheric Measurement Techniques, 2017_

## Referee Comment (RC1) · Anonymous Referee #1 · 30 Aug 2017

This is a short but important paper in that it describes the correction to OSIRIS pointing that is needed to make the data useful for trend calculation. The RSAS technique used to monitor the pointing errors is well established and they seem to understand its limitations. They have applied it to the OSIRIS data to determine a more accurate pointing and reprocessed the entire data set.

The correction for the long term trend in pointing, the goal of this exercise, is nicely done. Figure 5 shows this very clearly. But the extent to which the scatter is cleaned up, especially in the first two years is remarkable. Can you explain this reduction in scatter?

This paper could be published as is or with minor revisions.

minor comments and corrections:

Page 2 line 1 - need to note whether the trends Harris et al. refer to are for total column ozone or upper stratosphere.

P2 line 2 - "in upper stratospheric ozone..."

Figure 3 - a better color scale might show the RSAS offsets more clearly. With this scale you can't tell where the offsets are near zero.

Page 5 last paragraph - is the daily average correction latitude dependent? This needs to be described a bit more clearly.

It is probably noted in the Bourassa paper, but the pre 1997 trend from SAGE shown in Figure 7 has a comparison issue in the upper stratosphere because SAGE measures near terminator while OSIRIS measures at non-terminator times. This is not an issue for the MLS comparisons.

Need to say how the version 5.10 data can be obtained. Provide links please.

---

## Referee Comment (RC2) · Anonymous Referee #2 · 26 Sep 2017

**1   Summary**

This paper reports on additional corrections to the OSIRIS limb ozone retrieval that reduce an apparently too large upper stratospheric trends as reported by Bourassa et al. (2014). The correction is mainly an altitude registration correction called "RSAS" based on the analysis of the 350 nm limb radiance profile, which is also known as the "knee" method (utilising the information from a bend in the radiance profile), to derive a (constant) shift to be applied to the altitude coordinate prior to the trace gas retrieval. The main focus on the paper is to describe this correction procedure and in the second part to repeat the multilinear regression from Bourassa et al. (2014), but applied to the

corrected data, to update ozone profile trends up to year 2017.

The paper is very concise and well written and it is very suitable to AMT. I have one major issue which I think needs to be properly addressed before publication.

**2 Major issue**

This study assumes that the intial average and "stable" altitude bias of $+200$ m until $\sim$2011 as derived from RSAS is a modelling artifact (p.5,l.26) and then assumes that the true bias must be zero, thus offseting the entire time series of altitude shifts by -200 m (making the initial average bias zero). One argument in favor for doing this was that without altitude corrections the ozone bias from prior data versions with respect to other measurements (Adam's papers) was showing no meridional structure. I find this argument not convincing as we should not favor a certain procedure just to optimize a bias to some external data. It would be certainly better to show that indeed high level clouds introduce a systematic error in RSAS based upon RTM simulation. Alternatively, one could consider to show that by limiting OSIRIS data to cloud free scenes indeed a zero altitude bias can be found (at least on average). Figure 3 is in my regard only a hand waving argument.

**3 Minor comments**

p.1,l.23: In the list of references profile trend papers are cited, but the Weatherhead paper is a total ozone trend paper. Why is this paper cited?

p.2,l.8-10: Which of the three points raised by Harris et al. (2015) are addressed by this paper (probably i and ii), please state them.

[Figure]

p.4,l.25; change "350 km" to "350 nm".

p.4.,l.26: Why is the match done for radiances at 40 km and not higher? Does the albedo change if other altitudes are used? Please clarify.

p.4,l.27: omit "the scan is".

p.6,l.5: are " albedo, stratospheric aerosol extinction, NO2, and O3" retrieved simultaneously or is the same algorithm (with different settings) used to retrieve the various quantities separately. Please specify. You may want to add a reference here.

p.6,l.5-7: The sentence "No changes were made to the retrievals of the other species other than the pointing correction to the radiances; the impact of this correction on these other species has yet to be assessed." sounds awkward. What are "other species" than listed in the sentence before. You probably meant that the impact of the altitude corrections has only been assessed for ozone so far.

p.6,l.8: The sub phrase "is not equivalent to simply shifting the retrieved ozone profile by the RSAS correction" implies that somebody has used that simple solution. If yes, then please cite the study or explain where this has been done.

p.7,l.24ff: The trend results should be compared to more recent results from Steinbrecht et al. (2017) and Sofieva et al. (2017) that are also part of this special issue.

Figure 3: a better color scale for the contour plot that distinguishes between negative and positive values would be good.

Figure 4: Replace "GEOGRAPHIC AVERAGE FIGURE" in the caption by the correct figure number (guess Fig. 2).

---

## Author Comment (AC1) · 2 Nov 2017

**Author's Response:** amt-2017-229 originally submitted on 07 Jul 2017
**Title:** Drift corrected Odin-OSIRIS ozone product: algorithm and updated stratospheric ozone trends
**Authors:** A. Bourassa, C. Roth, D. Zawada, L. Rieger, C. McLinden, D. Degenstein

Thank you to both referees for the helpful reviews. The referee comments are provided below, with our response in blue.

**Referee #1:**

This is a short but important paper in that it describes the correction to OSIRIS pointing that is needed to make the data useful for trend calculation. The RSAS technique used to monitor the pointing errors is well established and they seem to understand its limitations. They have applied it to the OSIRIS data to determine a more accurate pointing and reprocessed the entire data set. The correction for the long term trend in pointing, the goal of this exercise, is nicely done. Figure 5 shows this very clearly. But the extent to which the scatter is cleaned up, especially in the first two years is remarkable. Can you explain this reduction in scatter?

We also agree that the reduction in scatter is quite remarkable, and attribute it to the robustness of the RSAS correction. There must exist short time scale (~daily) error in the Odin attitude solution that is reliably detected and corrected by the RSAS algorithm. As we note in the paper (section 3, paragraph 4): "The overall observed variability at shorter time scales has also decreased significantly in the v5.10 comparison. This effect can be observed in most time periods, but it is most noticeable from 2005 to mid-2007. The reduction in variability of this difference on a daily average scale gives us confidence that the short term variability in the calculated RSAS correction is in fact real and not simply noise or an artefact of the technique."

Page 2 line 1 - need to note whether the trends Harris et al. refer to are for total column ozone or upper stratosphere.

Yes, thank you. This statement refers to the upper stratosphere. We have clarified this in the revision.

P2 line 2 - "in upper stratospheric ozone..."

Correction done.

Figure 3 - a better color scale might show the RSAS offsets more clearly. With this scale you can't tell where the offsets are near zero.

A good point; we have revised the figure to use a more dynamic color scale that more clearly shows the lower values. See below.

Page 5 last paragraph - is the daily average correction latitude dependent? This needs to be described a bit more clearly.

The daily average correction is not latitude dependent, and we have now explicitly pointed this out in the text in this paragraph. We now state, "regions with high aerosol extinction or high/low albedo values that are discarded by the RSAS algorithm can have a large extent geographically, but even so there are still typically many valid scans in any given day that are used to construct the daily average. This yields one point per day with no geographical dependence." However, as noted in the last paragraph of section 4, "because the latitudinal coverage varies throughout the year due to the sun-synchronous terminator orbit,

the daily average RSAS pointing correction can manifest with meridional structure even though latitudinal dependence is averaged over any given day."

It is probably noted in the Bourassa paper, but the pre 1997 trend from SAGE shown in Figure 7 has a comparison issue in the upper stratosphere because SAGE measures near terminator while OSIRIS measures at non-terminator times. This is not an issue for the MLS comparisons.

Yes, the trend results must be treated carefully at the uppermost altitudes due to the photochemical dependence of ozone, although this is the case for essentially all trend analyses that use the SAGE II dataset, see for example the comprehensive results by Sofieva et al., AMT, 2017, and Steinbrecht et al., AMT, 2017.  As pointed out by the referee, this is not an issue for the MLS comparisons.

Need to say how the version 5.10 data can be obtained. Provide links please.

The link to the OSIRIS Level 2 products with download instructions is now provided.

[Figure]

**New version of** Figure 3: Globally binned in latitude and longitude RSAS offsets for time period beginning in 2004 and ending in 2010.  Values where the OSIRIS optics temperature is less than 18° C are excluded.

---

## Author Comment (AC2) · 2 Nov 2017

**Author's Response:** amt-2017-229 originally submitted on 07 Jul 2017
**Title:** Drift corrected Odin-OSIRIS ozone product: algorithm and updated stratospheric ozone trends
**Authors:** A. Bourassa, C. Roth, D. Zawada, L. Rieger, C. McLinden, D. Degenstein

Thank you to both referees for the helpful reviews. The referee comments are provided below, with our response in blue.

**Referee #2:**

Major issue: This study assumes that the initial average and "stable" altitude bias of +200 m until ~2011 as derived from RSAS is a modelling artifact (p.5,l.26) and then assumes that the true bias must be zero, thus offseting the entire time series of altitude shifts by -200 m (making the initial average bias zero). One argument in favor for doing this was that without altitude corrections the ozone bias from prior data versions with respect to other measurements (Adam's papers) was showing no meridional structure. I find this argument not convincing as we should not favor a certain procedure just to optimize a bias to some external data. It would be certainly better to show that indeed high level clouds introduce a systematic error in RSAS based upon RTM simulation. Alternatively, one could consider to show that by limiting OSIRIS data to cloud free scenes indeed a zero altitude bias can be found (at least on average). Figure 3 is in my regard only a hand waving argument.

This is a very good point and we have studied this in some detail. We have found that the RSAS correction over the 2004-2010 baseline time period is a monotonically increasing function of retrieved effective surface reflectance (albedo). This "effective surface" reflectance also incorporates the effect of clouds on the upwelling radiation. The mean RSAS offset over the 2004-2010 time period and associated standard deviation, is shown in a new figure (now labelled Figure 4 and shown below). For low values of albedo, the average RSAS offset is essentially zero (basically following the reviewer's suggestion to check the result for cloud-free scenes). The RSAS offset increases to approximately 400 m at high albedo. For a typical albedo of 0.3, the RSAS offset is ~200 m. This shows that for low values of albedo, the RTM simulation and thus the RSAS result is quite robust, and when the albedo increases, the uncertainty in the tropospheric conditions increases leading to a bias in the RTM simulation and the derived RSAS offset. This discussion has been added to the revised version of the paper.

Minor comments:

p.1,l.23: In the list of references profile trend papers are cited, but the Weatherhead paper is a total ozone trend paper. Why is this paper cited?

Citation removed.

p.2,l.8-10: Which of the three points raised by Harris et al. (2015) are addressed by this paper (probably i and ii), please state them.

Yes, we have added the statement that "this work contributes to the first two of these three points."

p.4,l.25; change "350 km" to "350 nm".

Done, thank you for catching this.

p.4.,l.26: Why is the match done for radiances at 40 km and not higher? Does the albedo change if other altitudes are used? Please clarify.

Increasing stray light and decreasing signal-to-noise limit the ability to use higher tangent altitudes. This statement has been added to the text.

p.4,l.27: omit "the scan is".

Done.

p.6,l.5: are " albedo, stratospheric aerosol extinction, NO2, and O3" retrieved simultaneously or is the same algorithm (with different settings) used to retrieve the various quantities separately. Please specify. You may want to add a reference here.

This is specified in Degenstein et al., 2009, which we now reference here.

p.6,l.5-7: The sentence "No changes were made to the retrievals of the other species other than the pointing correction to the radiances; the impact of this correction on these other species has yet to be assessed." sounds awkard. What are "other species" than listed in the sentence before. You probably meant that the impact of the altitude corrections has only been assessed for ozone so far.

Yes, the statement is now clarified to say exactly that: "The impact of the RSAS correction has only been assessed for ozone so far."

p.6,l.8: The sub phrase "is not equivalent to simply shifting the retrieved ozone profile by the RSAS correction" implies that somebody has used that simple solution. If yes, then please cite the study or explain where this has been done.

This has not been done with a published product to our knowledge; however, we thought it pertinent to point out since a simple shift of the retrieved profile might seem tempting to a naïve observer. In fact, we tried this with the OSIRIS data set and found that the non-linearity of the inversion does impact the results, especially at low altitude.

p.7,l.24ff: The trend results should be compared to more recent results from Steinbrecht et al. (2017) and Sofieva et al. (2017) that are also part of this special issue.

The trend results are very consistent with those presented by Steinbrecht et al., 2017, and Sofieva et al., 2017, which also use this drift-corrected OSIRIS data in combination with SAGE II, OMPS and other limb data records. This is noted in the text.

Figure 3: a better color scale for the contour plot that distinguishes between negative and positive values would be good.

Done. See response to referee #1.

Figure 4: Replace "GEOGRAPHIC AVERAGE FIGURE" in the caption by the correct figure number (guess Fig. 2).

Done. Yes, it's figure 2.

[Figure]

**New Figure 4: The dependence of the mean derived RSAS offset (and standard deviation) on retrieved effective surface reflectance over the same 2004-2010 time period.**